# Metagenomic Characterisation of the Gut Microbiome and Effect of Complementary Feeding on *Bifidobacterium* spp. in Australian Infants

**DOI:** 10.3390/microorganisms12010228

**Published:** 2024-01-22

**Authors:** Kimberley Parkin, Debra J. Palmer, Valerie Verhasselt, Nelly Amenyogbe, Matthew N. Cooper, Claus T. Christophersen, Susan L. Prescott, Desiree Silva, David Martino

**Affiliations:** 1Telethon Kids Institute, University of Western Australia, Nedlands, Perth, WA 6009, Australia; kimberley.parkin@telethonkids.org.au (K.P.);; 2Medical School, University of Western Australia, Nedlands, Perth, WA 6009, Australia; 3Larsson-Rosenquist Foundation Centre for Immunology and Breastfeeding, Medical School, University of Western Australia, Nedlands, Perth, WA 6009, Australia; 4School of Molecular Life Sciences, Curtin University, Bentley, Perth, WA 6102, Australia; 5School of Medical and Health Sciences, Edith Cowen University, Joondalup, Perth, WA 6027, Australia; 6Joondalup Health Campus, Joondalup, Perth, WA 6027, Australia; 7Nova Institute for Health, Baltimore, MD 21231, USA; 8Department of Family and Community Medicine, University of Maryland School of Medicine, Baltimore, MD 21201, USA; 9The ORIGINS Project, Telethon Kids Institute, Nedlands, Perth, WA 6009, Australia; 10School of Molecular Science, University of Western Australia, Nedlands, Perth, WA 6009, Australia

**Keywords:** gut microbiome, shotgun metagenomics, breastfeeding, *Bifidobacterium*, complementary feeding, the ORIGINS project

## Abstract

Complementary feeding induces dramatic ecological shifts in the infant gut microbiota toward more diverse compositions and functional metabolic capacities, with potential implications for immune and metabolic health. The aim of this study was to examine whether the age at which solid foods are introduced differentially affects the microbiota in predominantly breastfed infants compared with predominantly formula-fed infants. We performed whole-genome shotgun metagenomic sequencing of infant stool samples from a cohort of six-month-old Australian infants enrolled in a nested study within the ORIGINS Project longitudinal birth cohort. Infants born preterm or those who had been administered antibiotics since birth were excluded. The taxonomic composition was highly variable among individuals at this age. Predominantly formula-fed infants exhibited a higher microbiome diversity than predominantly breastfed infants. Among the predominantly breastfed infants, the introduction of solid foods prior to five months of age was associated with higher alpha diversity than solid food introduction after six months of age, primarily due to the loss of *Bifidobacterium infantis*. In contrast, the age at which solid food was introduced was not associated with the overall change in diversity among predominantly formula-fed infants but was associated with compositional changes in *Escherichia* abundance. Examining the functional capacity of the microbiota in relation to these changes, we found that the introduction of solid foods after six months of age was associated with elevated one-carbon compound metabolic pathways in both breastfed and formula-fed infants, although the specific metabolic sub-pathways differed, likely reflecting different taxonomic compositions. Our findings suggest that the age of commencement of solid foods influences the gut microbiota composition differently in predominantly breastfed infants than in predominantly formula-fed infants.

## 1. Introduction

The colonization of the gut microbiota during infancy is a critical period that has implications for infant health and development. Maternal and environmental factors, such as gestational age, mode of delivery, infant feeding, and exposure to antibiotics, can influence the specific bacterial taxa that sequentially colonize the gut and shape the overall composition and diversity [1]. Some studies have reported associations between infant microbiome development and the risk of certain diseases, such as atopic sensitization and allergic diseases [2,3,4,5]. A better understanding of the factors that shape early colonization patterns and the subsequent short- and long-term clinical consequences is needed to determine optimal health and inform practices targeting reduced disease risk.

Infant feeding is a major determinant of early microbiome composition, with infant gut microbes transmitted through breastfeeding among other body sites [6]. Breastfeeding promotes the colonization of *Bifidobacterium longum*, *Bifidobacterium breve*, and *Bifidobacterium bifidum*, which are well-adapted to human milk oligosaccharide (HMO) degradation [7]. Owing to the abundance of *Bifidobacterium* species, breastfeeding tends to maintain the gut in a state of lower diversity until cessation [8]. Formula milk feeding, on the other hand, is associated with a more diverse early gut ecosystem, typically characterized by additional species of the *Streptococcus* and *Enterococcus* genera [9]. The introduction of solid foods into the diet of infants is also associated with immense shifts in the diversity and taxonomic composition of the gut microbiome [10]. Such taxonomic changes are related to changes in the functional metabolic capacity of the microbiota, as solid foods generally promote increases in *Ruminococcus* and *Lactobacillus* species [11], which are highly specialized for carbohydrate digestion [9]. Despite dietary differences across geographically and culturally distinct populations, longitudinal data suggest that these transitions are widely observed and follow a similar progression in most infants [12].

In addition to feeding method, the timing of the introduction of solid foods has been studied for its influence on the gut microbiome diversity. Solid food introduction drives the infant gut microbiome into a more mature, adult-like state [13]. Earlier introduction of solid foods generally promotes an earlier increase in diversity due to the introduction of a wide variety of nutrients that are subsequently utilized as food sources by the gut microbiota [14]. However, there is some contention about whether breastfeeding duration is a more important determinant of early colonization patterns than the timing of solid food introduction [15]. In some studies, the effect of earlier introduction of solid foods on gut diversity appears to be independent of breastfeeding duration [14], while others suggest that breastfeeding duration, since it maintains a state of lower diversity, is more important for dictating early infant faecal microbiota profiles [16,17].

One aspect that has not been considered is whether the effects of the timing of solid food introduction into the infant diet have differential effects on the gut microbiota in breastfed versus formula-fed infants, which have very different baseline compositions at the time solid foods are introduced [9]. To explore this, we analysed shotgun metagenomic data from 170 six-month old infants from the ORIGINS Project, a Western Australian birth cohort.

## 2. Materials and Methods

### 2.1. Ethics Approval and Consent to Participate

All work in this study was completed in accordance with the ethical guidelines of the Ramsay Health Care (RHC) WA|SA HREC (EC00266; HREC reference number 1911). The participants’ mothers provided written informed consent on behalf of their children.

### 2.2. Study Participants

This study is a sub-project of The ORIGINS Project. This unique long-term study, a collaboration between Telethon Kids Institute and Joondalup Health Campus, is one of the most comprehensive studies of pregnant women and their families in Australia to date, recruiting 10,000 families over a decade from the Joondalup and Wanneroo communities of Western Australia [18]. Recruitment for this ORIGINS Project sub-study was conducted from November 2019 to March 2022, where mothers of participants were approached when their children were six months old. Inclusion criteria required infants to be healthy with no known congenital defects, born at ≥37 weeks of gestation, and to have had no antibiotics since birth.

### 2.3. Questionnaires and Operational Definitions

The mothers of the study participants completed web-based surveys when participants were six months of age. The questionnaires provided detailed information on participant demographics, and maternal and lifestyle factors. The majority of parents had already introduced solid foods to their infants’ diets at the time of sample collection. We defined ‘predominantly breastfed’ as those who reported consumption of less than or equal to one cup of infant formula per day on average in the first six months, and ‘predominantly formula-fed’ infants were those who reported consumption of less than or equal to one cup of breastmilk per day on average. The age of solid food introduction was stratified into three groups: <5 months of age, 5–6 months of age, and >6 months of age (no solids introduced at the time of sample collection), based on the earliest age at which any solid food was introduced. Solid food introduction was defined as any solids introduced, regardless of quantity.

### 2.4. Laboratory Processing and Bioinformatics Profiling

#### 2.4.1. Stool Sample Collection

Investigators provided mothers with stool sampling kits to collect stool samples from their six-month old’s soiled nappies. The stool samples were collected using a Copan FLOQSwab in an active drying tube (FLOQSwab-ADT, Copan Diagnostics, Murrieta, CA, USA). Stool sampling kits contained a pre-paid envelope to allow participants to ship their swabs directly to Microba Life Sciences Limited (Brisbane, Australia) a NATA-accredited facility. Upon arrival at the clinical laboratories, stool sample swabs were stored at −80 °C until DNA extraction and sequencing. DNA was subsequently extracted and subjected to shotgun metagenomic sequencing and microbiome profiling using standard workflows developed by Microba Life Sciences Limited.

#### 2.4.2. DNA Extraction

Faecal samples were extracted on the QIAcube HT system using the DNeasy 96 PowerSoil Pro QIAcube HT Kit (Qiagen 47021, Hilden, Germany), according to the manufacturer’s instructions, with a modified initial processing step (Qiagen 9001793). Mechanical lysis was performed using PowerBead Pro beads (Qiagen 19311). The resulting DNA was quantified using QuantIT, a high-sensitivity dsDNA fluorometric assay (Thermo Fisher, Q33120, Waltham, MA, USA). Samples were required to reach a minimum of 0.2 ng/μL to pass quality control requirements.

#### 2.4.3. Library Preparation

Libraries were constructed using the Illumina DNA Prep (M) Tagmentation Kit (Illumina 20018705, San Diego, CA, USA) and indexed with IDT for Illumina Nextera DNA Unique Dual Indexes Set A-D (Illumina 20027213-16) according to the manufacturer’s instructions, with a modification of the volume to accommodate processing in a 384-plate format. The resulting libraries were assessed using QuantIT (Thermo Fisher Q33120), and individual libraries were visualized by capillary gel electrophoresis using a QIAxcel DNA High Resolution Kit (Qiagen 929002).

#### 2.4.4. Sequencing

Individual libraries were pooled in equimolar amounts and assessed using QuantIT (Thermo Fisher Q33120) and visualized by capillary gel electrophoresis using a QIAxcel DNA High Resolution Kit (Qiagen 929002). Sequencing pools were loaded and sequenced on a NovaSeq6000 (Illumina) using v1.5 300 bp PE sequencing reagents according to the manufacturer’s instructions. The sequence data were reviewed to determine the general minimum performance requirements for yield and sequence quality.

#### 2.4.5. Metagenomic Sequencing Data Quality Control (QC)

Data quality control and processing were performed at Microba Life Sciences Limited. Paired-end DNA sequencing data were demultiplexed and adaptor-trimmed using the Illumina BaseSpace Bcl2fastq2 (v2.20), accepting one mismatch in the index sequences. Reads were then quality trimmed and residual adaptors were removed using the Trimmomatic v0.39 [19] software package with the following parameters: -phred33 LEADING:3 TRAILING:3 SLIDINGWINDOW:4:15 CROP:100000 HEADCROP:0 MINLEN:100. Human DNA was identified and removed by aligning reads to the human genome reference assembly 38 (GRCh38.p12, GCF_000001405) using bwa-mem v0.7.17 [20] with default parameters, except that the minimum seed length was set to 31 (-k31). Human genome alignments were filtered using SAMtools v1.7 [21], with flags -ubh-f1-F2304. Any read pairs in which at least one read mapped to the human genome with >95% identity and >90% of the read length were flagged as human DNA and removed. All samples were then randomly subsampled to a standard depth of seven million read pairs to standardize the number of reads across all samples to ensure they can be statistically compared.

#### 2.4.6. Quantification of Microbial Species, Gene, and Pathway Abundances

Species profiles were obtained with the Microba Community Profiler (MCP) v1.0 (www.microba.com, accessed October 2021) using the Microba Genome Database (MGDB) v1.0.3 as the genome reference database. Reads were assigned to genomes within the MGDB and the relative cellular abundance of species clusters was estimated and reported. Quantification of gene and pathway abundance in the metagenomic samples was performed using the Microba Gene and Pathway Profiler (MGPP) v1.0 against the Microba Genes (MGENES) database v1.0.3. MGPP is a two-step process. In step one, all open reading frames (ORFs) from all genomes in the MGDB were clustered against UniRef90 [22] release 2019/04 using 90% identity over 80% of the read length with MMSeqs2 Release 10-6d92c [23]. Enzyme Commission annotations were used to determine the encoding of MetaCyc [24] pathways in each genome using enrichment (https://github.com/geronimp/enrichM, accessed October 2021), and pathways that were complete or near-complete (completeness > 80%) were classified as encoded. In step two, all DNA sequencing read pairs aligned with one or more bases to the gene sequence from any protein within an MGENES protein cluster were summed. The abundance of encoded pathways of species reported as detected by MCP was calculated by averaging the read counts of all genes for each enzyme in that pathway.

### 2.5. Statistical Analyses

Analyses were performed using R version 4.1.3. All boxplots represent the first and third quartiles, with a median as a middle line and whiskers at the last value within a 1.5 × IQR distance from the upper or lower quartile, where IQR is the interquartile range. An alpha level 0.05 was used to define statistical significance for unadjusted and adjusted *p*-values as appropriate.

#### 2.5.1. Community Composition

The overall composition of the gut microbiome was visualized using taxonomic bar plots of the relative abundance of the top twelve most abundant species. Data were stratified by feeding method, and the average relative abundance composition was represented by the age at solid food introduction.

#### 2.5.2. Diversity Analysis

We calculated alpha and beta diversities using the R package phyloseq (version 1.38.0) [25]. Microbial diversity was estimated using abundance-dependent (Shannon) and -independent (Observed Richness) metrics to account for both richness and evenness in the data after rarefying the amplicon sequence variant (ASV) table to account for uneven sequencing depth. Data were stratified by feeding method and each group was rarefied to account for uneven sample depth. Whole cohort data were rarefied to 10,362,676 reads per sample, which is the sequencing depth of the sample with the lowest read count. We measured the alpha and beta diversity of count data among three different feeding methods with regard to milk: predominantly breastfed, predominantly formula-fed, or mixed-fed, delivery method, and age of solid food introduction. To test for alpha diversity differences, we used the Kruskal–Wallis test with Dunn’s post hoc test, and *p*-values were adjusted using Bonferroni correction. Beta diversity was calculated using the Bray–Curtis index and sample clustering was visualized using non-metric multidimensional scaling (NMDS). Differences in beta diversity between feeding methods were calculated using PERMANOVA [26].

#### 2.5.3. Differential Abundance

To identify ASVs and functional pathways that were differentially abundant between different ages of solid food introduction in breastfed and formula-fed infants, we used a non-parametric approach based on ranks of pairwise log ratio comparisons implemented in the R package ANCOMBC (version 2.2.1) [27]. ASVs present in at least 5% of the samples (9/170 samples) and with counts above 1000 were aggregated to the species rank. *p*-values were adjusted using the Benjamini–Hochberg correction. To further visualize specific differentially abundant species between breastfed and formula-fed infants, we created a volcano plot of the −log10 of each *p*-value and the log2 fold change, and we set a significance threshold of 0.05. We identified MetaCyc pathways and groups that were differentially abundant between infants with solids introduced <5 months and >6 months of age in breastfed and formula-fed infants. We used a model with a negative binomial distribution implemented in the R package DESeq2 (version 1.34.0) [28].

#### 2.5.4. Plots

We used functions within the R packages phyloseq (package version 1.38.0) [25], microViz (package version 0.10.0) [29], and ggplot2 (package version 3.4.0) [30] to create plots and visualize the data.

## 3. Results

### 3.1. Cohort Description

The general population characteristics are presented in Table 1. This study was nested within the ORIGINS Longitudinal Birth cohort [18]. A total of 170 full-term healthy infants provided stool samples for shotgun metagenomic sequencing. In our study population, 95 (55.9%) were born vaginally; in terms of their milk-based diet at six months of age, 93 (54.7%) were predominantly breastfed, 58 (34.1%) were predominantly formula-fed, and 19 (11.2%) were mixed-fed. Regarding the age of solid food introduction, 38 (22.4%) reported solid food introduction at <5 months of age, 114 (67%) reported solid food introduction between 5 and 6 months of age, and 12 (7%) reported no solid food introduction at the time of sample collection (>6 months of age). The general population characteristics stratified by age at solid food introduction are shown in Table 2. We identified balanced groups between milk-based diets at six months of age.

### 3.2. Metagenomic Characterisation of Faecal Samples

After quality filtering and downsampling, 2,147,751,354 reads were retained, with an average of 13,020,508 reads per sample (SD ± 345,168.5). An average of 77,046 human DNA reads (SD ± 874,288.8) were removed prior to analysis. All filtered reads were downsampled to at least 7,000,000 read pairs. One sample obtained only 6,676,182 read pairs but was still included in the analysis, as this sample was of high quality and the statistical power outweighed the minimal deviation. A total of 1305 bacterial ASVs belonging to 420 genera, 98 families, 42 orders, 17 classes, and 14 phyla were classified at the species level. Two ASVs belonged to the Archaea domain, *Methanobrevibacter_A smithii* and *Methanobrevibacter_A smithii_A*, and ten ASVs belonged to the Eukaryota domain: *Blastocystis* sp. *subtype 3*, *Blastocystis* sp. *subtype 4*, *Blastocystis* sp. *subtype 8*, *Candida albicans*, *Candida parapsilosis*, *Clavispora lusitaniae*, *Geotrichum candidum*, *Kluyveromyces marxianus*, *Saccharomyces group A*, and *Wickerhamiella pararugosa*.

### 3.3. Infant Gut Microbiome Taxonomy Is Highly Variable between Individuals and Primarily Defined by Feeding Method

We sought to describe the overall composition of the gut microbiome and to identify the most abundant taxa in our study population. Consistent with other reports in young infants, *Bifidobacterium infantis*, *Bifidobacterium longum*, and *Bifidobacterium breve* were the most abundant species in this population [31,32], although a large proportion of the cohort was dominated by taxa beyond the top twelve most abundant species (‘other’, Figure 1), highlighting the high rate of interindividual variability in our cohort (Figure 1A). Actinobacteria were highly variable in infants at six months of age, representing the dominant phyla in the majority of individuals, whereas others were dominated by Proteobacteria, Firmicutes_A, or Bacteroidota. Most infants were dominated by *Bifidobacterium* colonizing species, although some individuals were almost entirely colonized by *Escherichia*, or *Parabacteroides*.

Using multidimensional scaling analysis, we found that the type of milk-based diet was the largest contributor to variation in faecal communities (2.5% of the variance, PERMANOVA, *p* = 0.004) compared to the sex and delivery method (Figure 1B). This effect was clearly driven by the relative abundance of the *Bifidobacterium* genus, which exhibited a significant gradient effect across the different milk-based diet groups (9.9% of the variance, PERMANOVA, *p* = 0.001) (Figure 1C). We did not find any significant differences in alpha and beta diversities according to the mode of delivery in this age group (1.6% of the variance, PERMANOVA, *p* = 0.092). 

To understand whether the differences in community composition between feeding methods were influenced by the age of solid food introduction, we divided our data into predominantly breastfed and predominantly formula-fed infants in terms of their milk-based diet. We then identified the twelve most relatively abundant species and stratified them by age at solid food introduction. We found that the age at which solid food was introduced primarily affected the abundance of members of the *Bifidobacterium* genus. The most abundant species in the predominantly breastfed group, irrespective of the age at solid food introduction, were *Bifidobacterium infantis*, *Bifidobacterium breve*, *Bifidobacterium longum*, and *Bifidobacterium bifidum*. *Bifidobacterium infantis* and *Bifidobacterium breve* were more abundant in infants with solid food introduction after six months of age, and *Bifidobacterium bifidum* was more abundant in infants with solid food introduction before five months of age (Figure 1D).

In the predominantly formula-fed group, the most abundant species were *Bifidobacterium longum*, *Bifidobacterium breve*, and *Escherichia coli*. In this group, the abundance of members of the *Bifidobacterium* genus was affected by the age of solid food introduction. Solid food introduction after six months of age was associated with substantially more abundance of *Bifidobacterium longum* and *Bifidobacterium breve*, while *Bifidobacterium infantis* was more abundant in infants with solid food introduction before five months of age. Among predominantly formula-fed infants, the age of introduction of solid food influenced the emergence of *Escherichia* genus. Solid food introduction prior to five months of age was associated with a higher relative abundance of *Escherichia coli*, whereas solid food introduction after six months of age was associated with higher relative abundances of *Escherichia flexneri* (Figure 1E). Collectively, these data support the hypothesis that the timing of solid food introduction has different effects depending on the milk-based diet.

### 3.4. Age of Solid Food Introduction Influences Alpha Diversity in Breastfed Infants Only

To measure alpha diversity (the microbial diversity within a single sample), we used both observed richness (the number of distinct ASVs present) and the Shannon Index, which reflects both richness and evenness. Comparing the diversity of stool samples between feeding methods, we found that predominantly formula-fed infants had a higher mean number of colonizing species (richness) as well as a higher diversity across species compared with predominantly breastfed infants (*p* < 0.01) and mixed-fed infants (*p* = 0.021, and *p* = 0.0433) (Figure 2A). The Shannon Index measure of was the most variable in the predominantly breastfed group with a range of 0.28–3.05, 1.02–3.38 for the predominantly formula-fed group, and 1.17–3.02 for the mixed-fed group. As expected, in relation to the age of solid food introduction, introduction prior to 5 months of age was associated with higher Shannon Diversity scores in the entire cohort (*p* = 0.0299) (Figure 2B). We did not observe any significant differences between infants born using different delivery methods. As before, we stratified our data by a milk-based diet and compared the diversity measures across the age of the solid food introduction strata. In the predominantly breastfed group, solid food introduction before five months (richness, *p* = 0.047; Shannon Index, *p* = 0.019) and after six months (richness; *p* = 0.016, Shannon Index; *p* = 0.009) was associated with a higher richness and Shannon Index than introduction between five and six months (Figure 2C). We observed no significant difference in alpha diversity between the timing of solid food introduction in the predominantly formula-fed groups (Figure 2D).

### 3.5. Age of Solid Food Introduction Affects Differential Abundance of Different Species in Breast versus Formula-Fed Infants

Formal testing of species read counts was conducted using the ANCOMBC R package [27,33] to identify differentially abundant ASVs. We first subclassified our data into core microbial taxa, identified as taxa present in at least 5% of the samples. The taxa retained were then subjected to differential abundance analysis using the ANCOMBC package. We compared the timing of solid food introduction in predominantly breastfed and predominantly formula-fed infants. We identified 14 differentially abundant taxa in the predominantly breastfed group and 15 differentially abundant taxa in the predominantly formula-fed group. We found that solid food introduction prior to five months of age was associated with significantly (FDR < 0.05, log fold change ±2) lower *Bifidobacterium infantis* differential abundance among the predominantly breastfed group, consistent with the previous analysis, and *Clostridium paraputrificum,* a less abundant species (Figure 3A). Among the predominantly formula-fed group, the introduction of solid foods prior to 5 months of age was associated with higher colonization of *Escherichia coli* (Figure 3B).

### 3.6. Functional Potential

We assigned ASV reads to functional pathways annotated in the MetaCyc database [24] and compared the effect of the timing of solid food introduction on metabolic gene abundance between predominantly breastfed and predominantly formula-fed infants using DESeq2 testing [28]. A total of 901 functional pathways belonging to 51 groups were available for analysis. In both predominantly breastfed and the predominantly formula-fed infants, solid food introduction after six months modulated processes related to the breakdown and absorption of one-carbon compounds such as acetate and lactate (Figure 4A,B). Interestingly, among predominantly formula-fed infants, solid food introduction after six months was also associated with the up-regulation of processes related to the breakdown of chlorinated compounds that may be present in food in the form of pesticides, or in tap water in the form of chlorine disinfectants or chlorine disinfectant by-products [34]. At the individual pathway level, solid food introduction prior to five months of age was associated with the downregulation of different pathways in predominantly breastfed infants compared to predominantly formula-fed infants (Figure 4C,D). In both the predominantly breastfed and predominantly formula-fed infants, the majority of significant pathways were upregulated in the >6 months of age solid food introduction group. In the predominantly breastfed group, the most significantly upregulated group was PWY-5831~CDP-abequose biosynthesis, and in the predominantly formula-fed group, PWY-7178~ethylene glycol biosynthesis (engineered).

## 4. Discussion

The introduction of solid foods is a time of immense shift in the composition and functionality of the gut microbiota, driving the composition of the gut microbiome to a more adult-like state. However, the differing effects of solid food introduction between predominantly formula-fed and predominantly breastfed infants is largely unexplored. The major strength of this study was the deep metagenomic characterization of the gut microbiota at the species level in a well-characterized cohort of infants. We sequenced stool samples from 170 six-month-olds, the majority of whom had introduced solid foods into their diets, with a small group of infants having no solid food introduction. 

Consistent with previous studies, we identified that the overall taxonomy of the gut microbiome at six months of age is primarily defined by the feeding method [35]. In this study, we observed exceptionally high interindividual variation, consistent with this being a key period during which solid foods are introduced into the diet. Overall, Actinobacteria were identified as the dominant phylum, whereas specific individuals had little to no Actinobacteria and were colonized by Proteobacteria, Firmicutes_A, or Bacteroidetes. The abundance of Actinobacteria is largely a result of the high abundance of *Bifidobacterium* species, which are among the first microbes to colonize the infant gut because of their ability to digest HMOs. At the species level, the three most abundant species across the whole cohort were *Bifidobacterium infantis, Bifidobacterium longum*, and *Bifidobacterium breve*. However, some individuals did not carry these species. A loss of abundance of *Bifidobacterium* spp. in infants has been linked to an increased prevalence of obesity and metabolic disorders later in life [36]. Thus, understanding how the relationship between feeding method and age of solid food introduction affects *Bifidobacterium* abundance may lead to the development of appropriate feeding practices for optimal infant health.

Consistent with previous studies on infants in the first month of life [9,37,38], we found that a predominantly breastfed milk-based diet maintained a state of lower gut diversity, whereas the relative contributions of *Bifidobacterium* species were correlated with the age of solid food introduction. *Bifidobacterium infantis* was the most differentially abundant species influenced by the age of solid food introduction in predominantly breastfed infants. We also observed that the age of introduction of solid food influenced the abundance of *Clostridium paraputrificum,* a species involved in the production of short-chain fatty acids [39]. These trends were not entirely consistent among the predominantly formula-fed infants, who already exhibited a more diverse microbiota at six months of age.

Interestingly, the timing of solid food introduction in predominantly formula-fed infants had no effect on measures of diversity but certainly influenced the emergence of *Escherichia* spp. commensalism. These species play a role in the digestion of complex carbohydrates and proteins as well as in maintaining gut barrier integrity and reducing inflammation [40]. In predominantly breastfed infants, these commensals were present but at a much lower abundance. Therefore, our data suggest that *Bifidobacterium* spp. tend to outcompete other species in breastfed infants, but not in formula-fed infants. This indicates that the age of solid food introduction is a more important determinant of microbiome taxonomic composition than the feeding method [41]. We identified no differences in beta or alpha diversity between infants born via caesarean section and those born vaginally. Consistent with previous research, this suggests that significant differences between delivery methods in the infant gut microbiome disappear by six months of age [42].

The limitations of our study include a relatively small sample size, particularly in the mixed-fed infants and infants with solid foods introduced after six months of age. Future research with larger sample size could extend our conclusions in these groups. We were also limited in our ability to describe demographic factors that could have correlated with timing of solid food introduction, such as ethnicity, maternal age, and socioeconomic status. It is important to note that different formulations of infant formula may have different effects on the gut microbiome, particularly with some brands containing certain prebiotics, probiotics, or symbiotics. However, we were limited in our ability to examine this. We also lacked detailed information about the types and ranges of solid foods introduced to infants in our cohort. This information may be able to identify the components of infant formula or the specific food types that are most important in driving infant gut microbiome maturation. Sequencing breastmilk samples would also assist in identifying specific HMOs that uniquely affect *Bifidobacterium* abundance. The long-term effects of the age of solid food introduction between predominantly breastfed and predominantly formula-fed infants would require a follow-up stool sample after the cessation of a milk-based diet.

Functional analysis indicated that variations in the age at which solid foods were introduced into the diet influenced the metabolic pathways related to the utilization of one carbon (C1) compound as an energy source. We observed that this was consistent across breastfeeding and formula-fed infants. This reflects energy utilization from a wider variety of sources, including carbohydrates, fats, and proteins. Despite this, no solid food introduction at the time of sample collection was associated with different functional capacities between breastfed and formula-fed infants, likely reflecting the differences in taxonomic composition. Interestingly, in both predominantly breastfed and predominantly formula-fed infants, few pathways were significantly upregulated in infants with solid food introduced prior to five months of age. The only pathways significantly upregulated in infants with solid food introduction prior to five months of age were PWY-2721~trehalose degradation III in breastfed infants and P441-PWY~superpathway of N-acetylneuraminate degradation; all significantly differentially abundant pathways were upregulated in infants with no solid food introduction at the time of sample collection. This suggests that the metabolic profiles of infants are primarily defined by the age at solid food introduction rather than the feeding method.

There is some controversy in the literature regarding the diversity of the gut microbiome and disease prevention. A recent study by Ma et al. found that a higher diversity in wild-type mice was associated with an increased risk of allergy [43]. In contrast, another recent study by Hoskinson et al. found that delayed gut microbiome maturation was predictive of allergic disease later in life [44]. Thus, our study addresses a key gap in the literature on how maturation of the infant gut microbiome in relation to solid food introduction may depend on the feeding method. Our study has also provided novel insights into the unique effects of age at the introduction of solid foods on the microbiota of breast- and formula-fed infants.

## 5. Conclusions

The goal of this study was to characterize the six-month-old infant gut microbiome and determine whether the age of solid food introduction differentially affects the gut microbiome between predominantly breastfed and predominantly formula-fed infants. In doing so, we identified high rates of interindividual variation at six months of age. We identified feeding method as the key factor influencing community composition, primarily related to the abundance of *Bifidobacterium* species. We found that predominantly formula-fed infants had higher diversity than breastfed infants. Among predominantly breastfed infants, the introduction of solid foods prior to five months of age resulted in a more diverse gut microbiome. The age at solid food introduction was not associated with differences in diversity in predominantly formula-fed infants. Feeding method significantly affected the overall gut microbiome diversity, but the age at solid food introduction affected alpha diversity in breastfed infants only. We identified differentially abundant MetaCyc pathways between breastfed and formula-fed infants that were primarily related to one-carbon compound metabolism. We observed that the age of solid food introduction has differing effects depending on whether it is predominantly breastfed or a predominantly formula-fed diet.

## Figures and Tables

**Figure 1 microorganisms-12-00228-f001:**
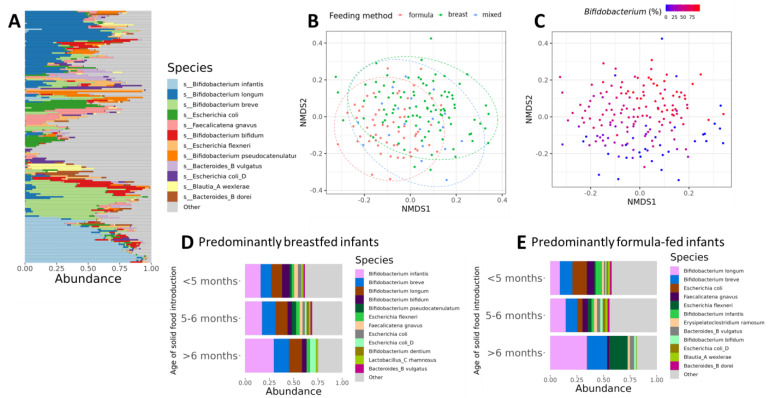
(**A**) Taxonomic bar plots showing 12 most abundant species sorted by Bray–Curtis dissimilarity for each participant. Y-axis is an individual coloured according to species relative abundance (x-axis). (**B**) Non-metric multidimensional scaling (NMDS) analysis of the gut microbiome community, with 95% data ellipses shown for each feeding method. Each point is an individual participant projected on the graph according to Bray–Curtis distance metric. (**C**) NMDS clustering with each individual coloured according to relative Bifidobacterium abundance. (**D**) Taxonomic bar plots for the top 12 most abundant species for the ‘predominantly breastfed’ group. Y-axis shows relative abundance expressed as a fraction. (**E**) Taxonomic bar plots for the top 12 most abundant species for the predominantly formula-fed group. Formula = predominantly formula-fed; breast = predominantly breastfed; mixed = mixed-fed.

**Figure 2 microorganisms-12-00228-f002:**
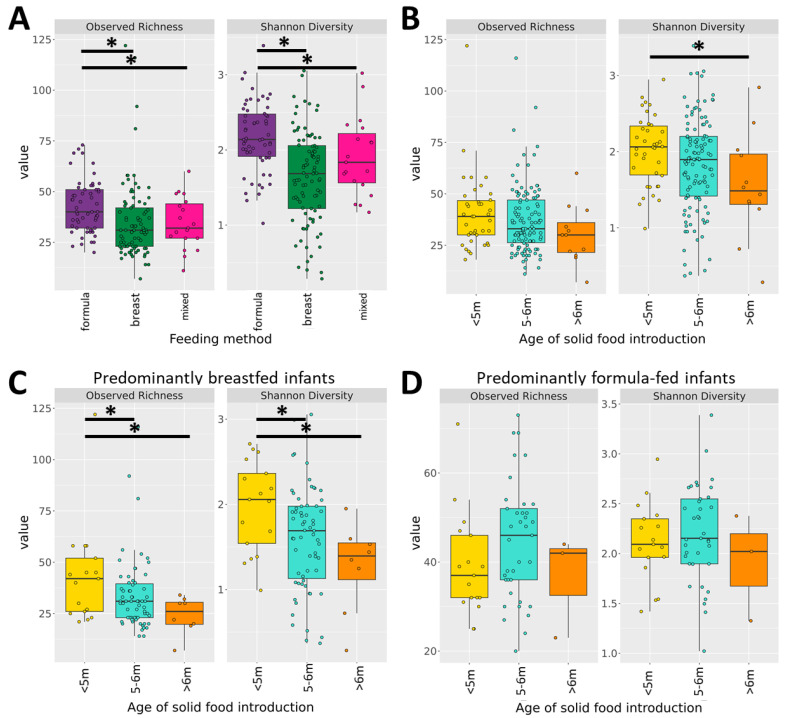
Alpha diversity distribution in considered groups in relation to (**A**) feeding method, and (**B**) age of solid food introduction. (**C**) Alpha diversity comparison between different ages of solid food introduction in breastfed infants. (**D**) Alpha diversity comparison between different ages of solid food introduction in formula-fed infants. Statistical significance (*p* < 0.05) indicated with an asterisk as assessed by the Kruskal–Wallis rank sum test. Box plots indicate medians with inter-quartile range. Formula = predominantly formula-fed; breast = predominantly breastfed; mixed = mixed-fed.

**Figure 3 microorganisms-12-00228-f003:**
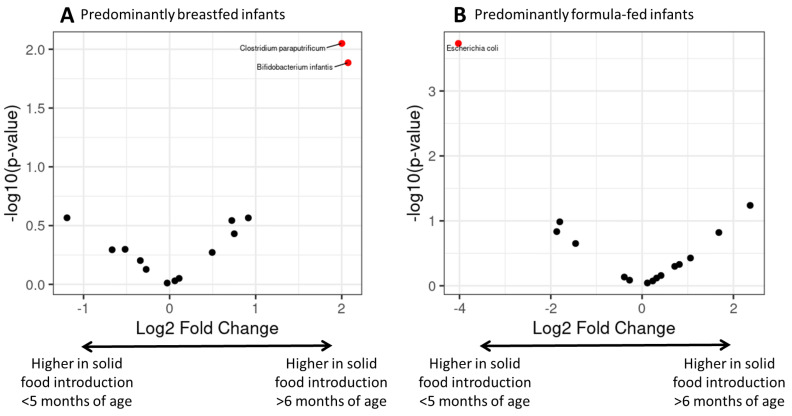
Volcano plots of differentially abundant ASVs between different ages of solid food introduction. (**A**) Predominantly breastfed group. Y-axis shows *p*-value for significance test and x-axis shows log fold change. (**B**) Predominantly formula-fed group. Each point represents a tested species, with significant species coloured red.

**Figure 4 microorganisms-12-00228-f004:**
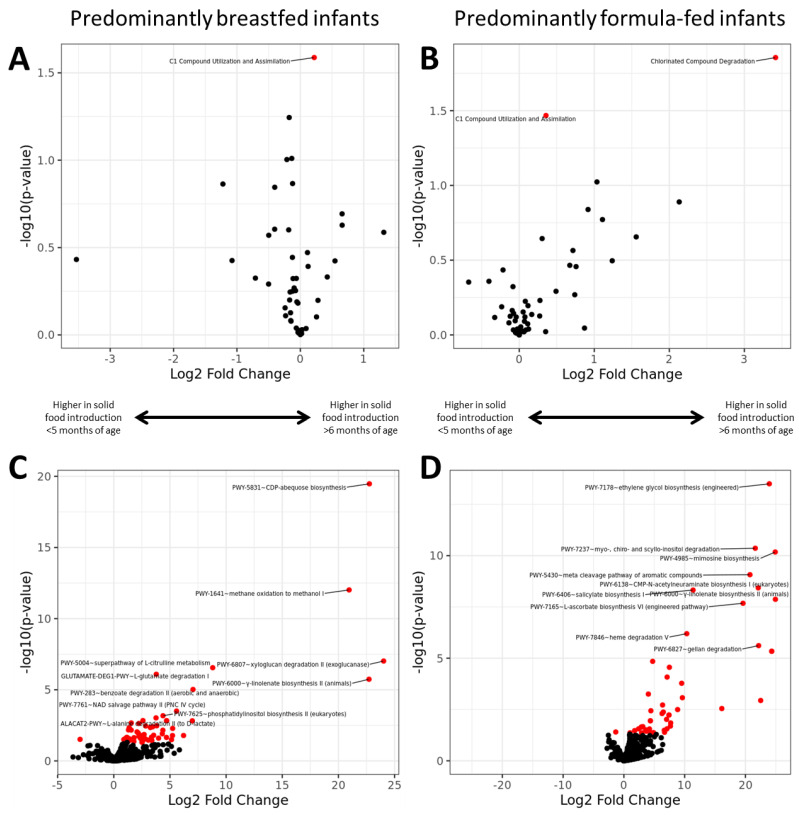
Differentially abundant MetaCyc metabolic groups and MetaCyc pathways between <5 months of age and >6 months of age solid food introduction groups. Results from analysis of MetaCyc metabolic groups comparison for (**A**) predominantly breastfed infants and (**B**) predominantly formula-fed infants. Results from analysis of MetaCyc pathways comparison for (**C**) predominantly breastfed infants and (**D**) predominantly formula-fed infants. Each point represents a differentially abundant ASV, with the ASVs coloured red passing a significance threshold of adjusted *p* < 0.05. The top 10 most significant MetaCyc pathways are labelled in (**C**,**D**).

**Table 1 microorganisms-12-00228-t001:** Summary of cohort characteristics and investigation variables.

	Level	n (%)	Missing Data (%)
n		170	
Sex	F	90 (52.9)	0 (0)
	M	80 (47.1)	
Milk-based diet at six months of age	Predominantly formula-fed	58 (34.1)	0 (0)
	Predominantly breastfed	93 (54.7)	
	Mixed-fed	19 (11.2)	
Delivery method	Emergency caesarean	35 (20.6)	0 (0)
	Planned caesarean	40 (23.5)	
	Vaginal	95 (55.9)	
Age of solid food introduction	<5 months	38 (22.4)	6 (3.5)
	5–6 months	114 (67)	
	>6 months	12 (7)	

**Table 2 microorganisms-12-00228-t002:** Summary of cohort characteristics stratified by age of solid food introduction. Percentages based on row totals.

		Age of Solid Food Introduction			
	Level	<5 Months of Age (%)	5–6 Months of Age (%)	>6 Months of Age (%)	*p*-Value
n		38	114	12	
Sex	F	13 (14.9)	68 (78.2)	6 (6.9)	0.024
	M	25 (32.5)	46 (59.7)	6 (7.8)	
Milk-based diet at six months of age	Predominantly formula-fed	17 (29.8)	37 (64.9)	3 (5.3)	0.595
	Predominantly breastfed	17 (19.3)	63 (71.6)	8 (9.1)	
	Mixed-fed	4 (21.0)	14 (73.7)	1 (5.3)	
Delivery method	Emergency caesarean	5 (15.2)	27 (81.8)	1 (3.0)	0.038
	Planned caesarean	9 (22.5)	24 (60.0)	7 (17.5)	
	Vaginal	24 (26.4)	63 (69.2)	4 (4.4)	

## Data Availability

Raw data supporting the conclusions of this article will be made available by the authors upon request.

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
