# Peer review of "Metagenomic Characterisation of the Gut Microbiome and Effect of Complementary Feeding on Bifidobacterium spp. in Australian Infants"

_microorganisms, 2024, doi:10.3390/microorganisms12010228_

Round 1

Reviewer 1 Report

Comments and Suggestions for Authors

The manuscript is well written and presents interesting data on the gut microbiome composition and functionality in 6-month-old infants based on their milk feeding type and time of introduction of solid foods.

I only have very minor comments on this manuscript:

1) what is the definition of introduction of solid foods? Was it any amount or for example at least 4 teaspoons of foods per day?

2) Stool sample collection: as written, it appears the stool samples remained at ambient temperature from collection from the nappy to posting to reception at the research centre. Can you confirm and/or indicate potential impact on the microbiota composition?

3 Line 260: I suggest to consider adding the following reference based on Australian infants, albeit at 2 months of age: doi: 10.1128/AEM.03910-12

4) line 328: please correct "give" for "five"

5) I invite the authors to consider the sample size, in particular for the mixed-fed group and introduction of solids after 6 months, as a limitation to the study

Author Response

R1.1) what is the definition of introduction of solid foods? Was it any amount or for example at least 4 teaspoons of foods per day?

RR 1.1 Thank you for bringing this point to our attention. We defined introduction of solid foods as a binary variable reflecting any amount of solid food introduced into the diet in the time covered by the questionnaire.

In the revised manuscript, we have clarified this by including the following statement in the methods (line 112): “Solid food introduction was defined as any solids introduced, regardless of quantity” 

R1.2) Stool sample collection: as written, it appears the stool samples remained at ambient temperature from collection from the nappy to posting to reception at the research centre. Can you confirm and/or indicate potential impact on the microbiota composition?

RR 1.2. Thank you for this question. Stool samples were collected using FLOQSwab in an active drying tube (FLOQSwab-ADT), which has been thoroughly evaluated against other gold standard preservatives over a range of temperatures by our colleagues at Microba LifeSciences (https://pubmed.ncbi.nlm.nih.gov/37938632/). This method was found to outperform others for up to four weeks, so we expect negligible impact on the microbial composition over the few days of transit between collection site and the extraction lab.

R1.3 Line 260: I suggest to consider adding the following reference based on Australian infants, albeit at 2 months of age: doi: 10.1128/AEM.03910-12

RR 1.3. Thank you, in the revision we have now added this reference which is cited on line 261 of the main text.

R1.4) line 328: please correct "give" for "five"

RR 1.4. Typo corrected. Sentence now reads, “…a higher richness and Shannon Index than introduction between five and six months”.

R1.5) I invite the authors to consider the sample size, in particular for the mixed-fed group and introduction of solids after 6 months, as a limitation to the study

RR1.5. Thankyou we agree with this suggestion. We have added the following text to the discussion on line 431:

“Limitations of our study include a relatively small sample size, particularly in the mixed-fed infants and infants with solid foods introduced after six months of age. Future research with larger sample size could extend our conclusions in these groups.”

Reviewer 2 Report

Comments and Suggestions for Authors

The manuscript by Kimberley Parkin and colleagues is titled "Metagenomic characterization of the gut microbiome and the effects of complementary feeding on Bifidobacterium spp. in Australian infants". Supplemental feeding can induce dramatic ecological shifts in infant gut microbiota toward more diverse compositions and functional metabolic capacities, which could have significant implications for immune and metabolic health. This study aimed to determine whether the age at which solid foods are introduced differentially affects the microbiota of predominantly breastfed infants when compared to predominantly formula-fed infants. A cohort of six-month-old Australian infants enrolled in a nested study within the ORIGINS Project longitudinal birth cohort was subjected to whole-genome shotgun metagenomic sequencing.Preterm babies and infants who have been administered antibiotics since birth were excluded from the study. At this age, the taxonomic composition of individuals was highly variable. Microbiomes of predominantly formula-fed infants are more diverse than those of predominantly breastfed infants. The introduction of solid foods prior to five months of age was associated with a higher alpha diversity than the introduction of solid foods after six months of age, primarily due to the loss of Bifidobacterium infantis. Among predominantly formula-fed infants, the age at which solid food was introduced was not associated with the overall change in diversity, however, it was associated with compositional changes in Escherichia abundance. When examining the functional capacity of the microbiota in relation to these changes, we determined that the introduction of solid foods after the age of six months led to elevated 1-carbon compound metabolic pathways in breastfed infants and formula-fed infants, although the specific metabolic sub-pathways differed, likely reflecting differing taxonomic compositions. It appears that the age of introduction of solid foods influences the composition of the gut microbiota differently in predominantly breastfed infants compared to predominantly formula fed infants. I have a few comments regarding the present manuscript.

  • The manuscript appears to be well-written, and the objectives are clearly stated and the aim is well presented

  • Although the metagenomic approaches are different from my own laboratory procedures, they are well explained and cited in the material and methods.

  • Can you add novel analyses such as LefSe analysis or PICRUSt to complete your snapshot of 170 participants? 

  • Are there any other variables that could be shown to the readers in order to enhance the present manuscript?

Author Response

R2.1 Can you add novel analyses such as LefSe analysis or PICRUSt to complete your snapshot of 170 participants? 

RR2.1 We sincerely appreciate these valuable suggestions for enriching our investigation of this cohort. We fully agree that both LefSe and PICRUSt are powerful tools for functional inference and could potentially offer additional insights.

However, after careful consideration, we believe that incorporating these analyses at this stage are unlikely to significantly alter the core conclusions of our study, considering the comprehensive analyses already conducted using ANCOM-BC and enrichM. Our selected methods align well with established practices in the field and yielded results consistent with previous studies for our data set.

Implementing LefSe and PICRUSt in our current workflow would require substantial additional time and computational resources, potentially hindering the timely publication of our findings.

Nevertheless, we acknowledge the valuable potential of these tools and plan to utilize them in future studies.

R2.2 Are there any other variables that could be shown to the readers in order to enhance the present manuscript?

RR 2.2. The reviewer raises an excellent point about enriching the manuscript by exploring additional variables potentially linked to the timing of solid food introduction. We readily acknowledge the value of considering factors like ethnic background, maternal age, and socioeconomic status in this context.

Half of our cohort were enrolled into the ORIGINS parent project as ‘non-active’ participants, meaning they consented only to provide passive surveillance data (which does not include a number of these potentially interesting additional factors). Due to the high rate of missing data for additional demographic variables we opted only to include variables with complete data.

In the revised manuscript we have acknowledged this limitation on line 433 of the discussion:

“We were also limited in our ability to describe demographic factors that could have correlated with timing of solid food introduction, such as ethnicity, maternal age, and socioeconomic status”

Round 2

Reviewer 2 Report

Comments and Suggestions for Authors

Thank you for taking into account my previous comments. Thank you again for clarifying the procedure for obtaining the information. I have no further comments to make.

Author Response

We'd like the thank the reviewer for their feedback and comments.